# Neonatal morbidity after fetal exposure to antipsychotics: a national register-based study

Essi Heinonen ,[1,2] Lisa Forsberg,[1] Ulrika Nörby,[3,4] Katarina Wide,[1] Karin Källén[3]

[1]Department of Clinical Sciences Intervention and Technology, Karolinska Institutet, Stockholm, Sweden
[2]Department of Neonatology, Karolinska University Hospital, Stockholm, Sweden
[3]Department of Clinical Sciences, Centre of Reproduction Epidemiology, Tornblad Institute, Lund University, Lund, Sweden
[4]Health and Medical Care Administration, Region Stockholm, Stockholm, Sweden

**Correspondence to**
Essi Heinonen;
essi.heinonen@ki.se

## ABSTRACT

**Objective** To investigate the admission rate to neonatal care and neonatal morbidity after maternal use of antipsychotics during pregnancy.

**Design** A population-based register study.

**Setting** Information on all singleton births between July 2006 and December 2017 in Sweden including data on prescription drugs, deliveries and infants' health was obtained from the Swedish Medical Birth Register, the Prescribed Drug Register and the Swedish Neonatal Quality Register. Exposed infants were compared with unexposed infants and with infants to mothers treated with antipsychotics before or after but not during pregnancy.

**Participants** The cohort comprised a total of 1 307 487 infants, of whom 2677 (0.2%) were exposed to antipsychotics during pregnancy and 34 492 (2.6%) had mothers who were treated before/after the pregnancy.

**Outcome measures** The primary outcome was admission rate to neonatal care. Secondary outcomes were the separate neonatal morbidities.

**Results** Of the exposed infants, 516 (19.3%) were admitted to neonatal care compared with 98 976 (7.8%) of the unexposed infants (adjusted risk ratio (aRR): 1.7; 95% CI: 1.6 to 1.8), with a further increased risk after exposure in late pregnancy. The highest relative risks were seen for withdrawal symptoms (aRR: 17.7; 95% CI: 9.6 to 32.6), neurological disorders (aRR: 3.4; 95% CI: 2.4 to 5.7) and persistent pulmonary hypertension (aRR: 2.1; 95% CI: 1.4 to 3.1) when compared with unexposed infants. The absolute risks for these outcomes were however low among the exposed infants, 1.3%, 1.8% and 1.0%, respectively, and the relative risks were lower when compared with infants to mothers treated before/after the pregnancy.

**Conclusion** Fetal exposure to antipsychotics was associated with an increased risk of neonatal morbidity. The effects in the exposed infants seem transient and predominantly mild, and these findings do not warrant discontinuation of a necessary treatment but rather increased monitoring of these infants. The increased risk of persistent pulmonary hypertension requires further studies.

## INTRODUCTION

Use of antipsychotic drugs is increasing globally due to the widened indications and off-label use of second-generation antipsychotics in treatment of a wide range of psychiatric

## STRENGTHS AND LIMITATIONS OF THIS STUDY

⇒ To our knowledge, this is the largest study to date to investigate the risk of individual neonatal morbidities among infants exposed to antipsychotics.
⇒ The Swedish health registers have the advantage of almost complete coverage of the inhabitants, information on important confounders and minimal bias in data collection, with the Swedish Neonatal Quality Register containing detailed information on all infants admitted to neonatal care.
⇒ We included a control group of infants to mothers treated with antipsychotics before or after but not during pregnancy that allowed us to adjust for factors associated with the underlying disorder, at least to some extent.
⇒ Even though the study included 1.3 million births, the numbers were too low to obtain reliable risk estimates for specific antipsychotics. Instead, the analyses were limited to antipsychotics on group level.

disorders, also during pregnancy.[1–6] Treatment during pregnancy does not generally seem to increase the risk of malformations.[7–12] Some studies have indicated that schizophrenia itself might be connected to an increased risk of malformations, and discontinuation of drug therapy is not recommended during pregnancy due to the risk of relapse of the underlying condition.[13–18]

Fetal exposure to antipsychotics has in smaller cohorts been linked to increased risks of prematurity, altered fetal growth, increased admission to neonatal intensive care unit (NICU), low APGAR scores, and neonatal extrapyramidal and respiratory symptoms. These risks seem to increase with increased dosage and with polypharmacy. Several of the previous studies have not adjusted the risks for any confounding maternal factors.[9 12 16 19–21] Three studies that adjusted for health and lifestyle confounders in the analyses did not confirm the increased neonatal morbidity after intrauterine antipsychotic exposure. Instead, one of them found a correlation between maternal psychiatric care during

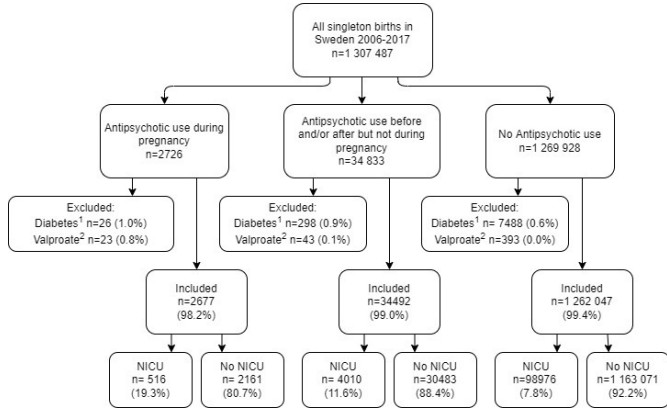

**Figure 1** Flow chart over the study design. [1]Women with pre-pregnancy diabetes mellitus. [2]Use of valproate during pregnancy. NICU, neonatal intensive care unit.

pregnancy and infant admission to NICU.[22–24] Maternal schizophrenia and bipolar disorders have, independent of drug exposure, been associated with increased risks of prematurity, pre-eclampsia, neonatal morbidity including poor arousal and potential adverse neurodevelopmental effects in the infants.[3 14 25–27] Mothers with untreated psychiatric illness are also more likely to engage in high-risk behaviours such as smoking, alcohol and illicit drugs and are less likely to attend to antenatal care.[25]

The aim of this Swedish population-based register study was to clarify the pattern and the frequency of the neonatal effects associated with intrauterine antipsychotic exposure. We also attempted to separate these effects from the impact of the underlying disease, its associated risk factors and other medications.

## METHODS

The study was a register-based study combining data from the Swedish Medical Birth Register (MBR),[28] the Prescribed Drug Register (PDR),[29] the Swedish Neonatal Quality Register (SNQ)[30] and the Perinatal Revision South Register (PRS).[31] Swedish personal identification numbers were used for register linkages. The study population consisted of all singleton births in Sweden, a total of 1 307 487 infants, registered in the MBR between 1 July 2006 and 31 December 2017. An outline of the study design is presented in figure 1.

The MBR holds data on antenatal care, delivery and examination of the newborn for >97% of all births. Information on body mass index (BMI), medications and smoking habits is registered in the MBR at the first visit of the complimentary antenatal care offered to all pregnant women in Sweden.[28] For this study, information on drug exposure and maternal and fetal background characteristics was collected from the MBR. Women with a diagnosis of pre-pregnancy diabetes or with valproate treatment (N03AG01) were excluded from the analysis. Valproate-exposed infants were excluded due to the well-known teratogenicity of the drug as well as co-medication with valproate and antipsychotics being fairly common.[32]

Data on exposure for prescription drugs were also acquired from the PDR. The PDR stores data on all drugs prescribed in ambulatory care and dispensed at a Swedish pharmacy but does not include medications used in hospitals.[29] The drugs registered in the MBR and PDR are classified according to the Anatomical Therapeutic Chemical (ATC) classification system. Antipsychotic exposure was defined as drugs belonging to ATC-class N05A, antipsychotics. Exposures to the antipsychotics dixyrazine (N05AB01), prochlorperazine (N05AB04), melperone (N05AD03) and lithium (N05AN01) were excluded from the exposed group and considered as covariates, due to the use of these drugs as antiemetics (dixyrazine, prochlorperazine, melperone) and as a mood stabiliser (lithium) rather than antipsychotics. The remaining drugs in class N05A were divided into first-generation and second-generation antipsychotics according to online supplemental table 1.

Antipsychotics exposure was allocated into any exposure (drugs dispensed at any time during or 1 month before the pregnancy), late exposure (drugs dispensed during the last 90 days of the pregnancy with or without earlier dispenses) and early exposure only (drugs dispensed 1 month before and during pregnancy but not during the last 90 days of the pregnancy). We also created a reference group with women exposed to antipsychotics anytime during the study period, before or after the pregnancy, but not during the pregnancy or 1 month before it. This group was used to attempt to control for the effect of the underlying psychiatric illness and other psychosocial factors connected to the neonatal outcome. Exposure data were also collected on the following neurotropic drugs known to or suspected to cause similar neonatal morbidity as antipsychotics: antidepressants (ATC-code N06A), antiepileptics (N03A), opioids (N02A), centrally acting sympathomimetics (N06BA), sedatives (N05B, N05C) and milder sedatives (alimemazine, promethazine and the excluded antiemetic antipsychotics from N05A). Concomitant neurotropic drug exposure was defined as a prescription of these drugs dispensed at any time during or 1 month before the pregnancy and was adjusted for in the multivariable regression model.

Data on NICU admissions and neonatal outcomes were extracted from the SNQ that covers all 37 Swedish NICUs.[30] Because the south of Sweden was not included in the SNQ until 2012, data from this region were collected from the PRS from 2006 to 2011.[31] Both registers comprise detailed information on infants treated at neonatal wards. The infants' diagnoses are registered in the SNQ/PRS and MBR according to the International Classification of Diseases, 10th Edition (ICD-10), or as checkboxes in the infant's medical record for some diagnoses in the SNQ. The care at NICUs in Sweden corresponds to the American Academy of Paediatrics' classifications of neonatal care, levels II–IV.[33] Infants with minor neonatal complications may remain in the maternity ward (equivalent to level I care) and are not included in the SNQ or PRS. Their diagnoses are, however, registered in the

MBR. Infant birth weight z-scores were estimated based on infant weight for gestational age and sex according to a Swedish ultrasound-based intrauterine growth curve.[34]

We used the Strengthening the Reporting of Observational Studies in Epidemiology cohort checklist when writing our report.[35]

## Patient and public involvement

No patient involved.

## Statistical analyses

Exposures were any antipsychotic use versus no use, use of the different antipsychotic groups versus no use, use of antipsychotics in early and late pregnancy, respectively, versus no use, and use during versus use before or after pregnancy. Outcomes (with ICD-10-codes in parentheses) were admission to NICU (yes/no, duration), transient tachypnoea of the newborn (P221, P228 or checkbox in SNQ), persistent pulmonary hypertension (PPHN) (P293B or checkbox in SNQ), respiratory distress syndrome (P220), hyperbilirubinaemia (P58, P59 or checkbox in SNQ), hypoglycaemia (P704 or checkbox in SNQ), feeding difficulties (P92 or checkbox in SNQ), neurological disorders (P909 seizures, P941-42 congenital hyper/hypotonia, P910-9 disturbances of cerebral status or checkbox in SNQ), withdrawal symptoms (P961-2), any malformations (all Q-codes with exception for Q18, Q65, Q270, Q250 in term infants or Q53 in term infants), heart malformations (Q20) and need for treatment with continuous positive airway pressure (DG001) or ventilator (DG002). Risk ratios (RRs) for dichotomous outcomes were obtained by using modified Poisson regression in multivariable regression models. Crude and adjusted RRs are displayed. In the final analyses, adjustments were made for: maternal age (continuous), primiparity (vs multiparity), maternal smoking (ordinal: 1=no smoking, 2=smoking <10 cigarettes/day, 3=smoking ≥10 cigarettes/day, entered as a continuous variable), BMI (continuous), maternal use of other neurotropic drugs (listed in table 1) and caesarean section (vs vaginal birth). As a sensitivity analysis, the risks were also adjusted for gestational age and z-score for birth weight to examine how much of the neonatal morbidity could be attributed to these factors (online supplemental tables 2 and 3). Missing data regarding maternal smoking and BMI were low and replaced by the overall means (table 1). For descriptive data, $X^2$ tests were used to detect heterogeneity between exposure groups (table 1).

Length of stay at neonatal ward is presented as medians and IQRs for the three exposure groups. Difference in the length of stay at neonatal ward between exposed and never exposed infants was evaluated by using univariate analysis of variance of the logarithmic variable for length of stay that followed the normal distribution. Number needed to harm (NNH) was calculated from the adjusted risk difference between exposed and never exposed infants. For this analysis, the adjusted frequencies of NICU admission, PPHN and neurological disorders among the exposed infants were calculated from the results of the multivariate risk analyses for these outcomes, to adjust for the risk of the underlying maternal condition. The statistical analyses were conducted by using SPSS V.27 (IBM SPSS Statistics).

## RESULTS

Among the included infants, 2677 (0.2%) were exposed to antipsychotics during pregnancy, and 34 492 (2.6%) had mothers treated with antipsychotics before or after but not during the pregnancy. The study outline is presented as a flow chart in figure 1. The most prescribed antipsychotics in the pregnant population during the study period were quetiapine (1021 exposures), olanzapine (771 exposures) and aripiprazole (334 exposures), all second-generation antipsychotics. The most prescribed first-generation antipsychotic was levomepromazine with 251 exposures (online supplemental table 1).

Table 1 summarises the background characteristics of the included infants and their mothers. The infants exposed to antipsychotics were more likely to be born with a caesarean section, to be small for gestational age and to be born preterm, and their mothers were more likely to smoke, be overweight or obese, not be living with the father of the child and to be using concomitant neurotropic medications, compared with both unexposed infants and infants to mothers treated with antipsychotics before and/or after the pregnancy. In the exposed group, 40.7% of the mothers used any other neurotropic drug listed in table 1 during pregnancy, compared with 2.7% in the unexposed group. No statistically significant difference between the groups was found in the incidence of pre-eclampsia, the infant being large for gestational age, perinatal death or infant sex.

The risk of being admitted to neonatal care was increased for infants exposed to antipsychotics in early pregnancy only compared with non-exposed peers (adjusted RR (aRR): 1.5, 95% CI: 1.3 to 1.7). Exposure in late pregnancy increased the risk further (aRR: 1.8, 95% CI: 1.6 to 2.0), adjusted for concomitant use of other psychotropic drugs. In sensitivity analyses, the relative risk for neonatal care admission for infants exposed to antipsychotics during late pregnancy versus non-exposed was instead stratified by use of other psychotropic drugs and weas found to be similar to the previously obtained aRRs. The aRRs were 1.7 (95% CI: 1.4 to 2.0) and 2.0 (95% CI: 1.8 to 2.3) for children exposed or not exposed to other psychotropic drugs, respectively. The risk of being admitted to NICU was similar for infants exposed to first-generation and second-generation antipsychotics (table 2). When compared with infants born to women treated with antipsychotics before or after but not during pregnancy, the risk increase for admission to NICU was lower but still statistically significant. Additional adjustment for gestational age and z-score for infant weight did not markedly reduce the risk for being admitted to NICU for any group of infants (online supplemental table 2).

**Table 1** Background characteristics of the study population

| | Antipsychotics during pregnancy n=2677 | Antipsychotics before or after but not during pregnancy n=34 492 | No antipsychotic use* n=1 262 047 | Antipsychotics during pregnancy vs antipsychotics before/after pregnancy | Antipsychotics during pregnancy vs no antipsychotic use | Antipsychotics before/after pregnancy vs no antipsychotic use |
|---|---|---|---|---|---|---|
| | n (%) | n (%) | n (%) | P value | P value | P value |
| Year of child birth | | | | <0.001 | <0.001 | <0.001 |
| 2006–2010 | 1013 (37.8) | 19 279 (55.9) | 611 663 (48.3) | | | |
| 2011–2017 | 1664 (62.2) | 15 213 (44.1) | 650 384 (51.5) | | | |
| Maternal age, years | | | | <0.001 | 0.020 | <0.001 |
| <20 | 70 (2.6) | 1059 (3.1) | 25 554 (2.0) | | | |
| 20–35 | 1883 (70.3) | 26 638 (77.2) | 964 380 (76.4) | | | |
| 35+ | 724 (27) | 6795 (19.7) | 272 113 (21.6) | | | |
| Parity | | | | <0.001 | <0.001 | <0.001 |
| Primipara | 1891 (70.6) | 20 727 (60.1) | 797 603 (63.2) | | | |
| Multipara | 786 (29.4) | 13 764 (39.9) | 456 556 (36.2) | | | |
| BMI in early pregnancy | | | | <0.001 | <0.001 | <0.001 |
| <24.9 | 1066 (39.8) | 17 284 (50.1) | 717 953 (56.8) | | | |
| 25–29.9 | 772 (28.8) | 8516 (24.7) | 298 162 (23.6) | | | |
| ≥30 | 618 (23.1) | 5723 (16.6) | 147 627 (11.7) | | | |
| BMI unknown | 221 (8.3) | 2969 (8.6) | 98 302 (7.8) | | | |
| Smoking in early pregnancy | | | | <0.001 | <0.001 | <0.001 |
| No | 1818 (67.9) | 26 774 (77.6) | 1 128 043 (89.4) | | | |
| Yes | 733 (27.4) | 5939 (17.2) | 67 348 (5.3) | | | |
| Missing information | 126 (4.7) | 1779 (5.2) | 66 656 (5.3) | | | |
| Maternal country of birth | | | | <0.001 | 0.209 | <0.001 |
| Sweden | 2018 (75.4) | 27 655 (80.2) | 928 818 (73.6) | | | |
| Other Nordic | 123 (4.6) | 946 (2.7) | 51 223 (4.1) | | | |
| Non-Nordic | 525 (19.6) | 5844 (16.9) | 257 259 (20.4) | | | |
| Maternal cohabitation | | | | <0.001 | <0.001 | <0.001 |
| Not living with father of child | 657 (24.5) | 4824 (14.0) | 75 451 (6.0) | | | |
| Concomitant neurotropic drugs | | | | | | |
| Lithium (N05AN01) | 124 (4.6) | 292 (0.8) | 2 (0.0) | <0.001 | <0.001 | <0.001 |
| Opioids (N02A) | 44 (1.6) | 604 (1.8) | 4748 (0.4) | 0.680 | <0.001 | <0.001 |
| Antiepileptics (N03A) | 215 (8.0) | 725 (2.1) | 2989 (0.2) | <0.001 | <0.001 | <0.001 |
| Antidepressants (N06A) | 755 (28.2) | 4409 (12.8) | 23 926 (1.9) | <0.001 | <0.001 | <0.001 |
| Psychostimulants (N06B) | 88 (3.3) | 328 (1.0) | 790 (0.1) | <0.001 | <0.001 | <0.001 |
| Anxiolytics and sedatives (N05B, C) | 436 (16.3) | 1429 (4.1) | 4620 (0.4) | <0.001 | <0.001 | <0.001 |
| Milder anxiolytics and antiemetics† | 366 (13.7) | 2516 (7.3) | 24 422 (1.0) | <0.001 | <0.001 | <0.001 |
| Pregnancy complications | | | | | | |
| Gestational diabetes | 76 (2.8) | 493 (1.4) | 14 979 (1.2) | <0.001 | <0.001 | <0.001 |
| Pre-eclampsia | 25 (0.9) | 346 (1.0) | 10 345 (0.8) | 0.729 | 0.512 | <0.001 |
| Caesarean section | 687 (25.7) | 7178 (20.8) | 206 586 (16.4) | <0.001 | <0.001 | <0.001 |
| Preterm birth | 259 (9.7) | 2471 (7.2) | 67 637 (5.3) | <0.001 | <0.001 | <0.001 |
| Small for gestational age | 104 (3.9) | 929 (2.7) | 29 182 (2.3) | <0.001 | <0.001 | <0.001 |
| Large for gestational age | 126 (4.7) | 1410 (4.1) | 50 421 (4.0) | 0.121 | 0.060 | 0.386 |
| APGAR 5 min <7 | 90 (3.4) | 737 (2.1) | 17 244 (1.4) | <0.001 | <0.001 | <0.001 |

Continued

**Table 1** Continued

| | Antipsychotics during pregnancy n=2677 | Antipsychotics before or after but not during pregnancy n=34 492 | No antipsychotic use* n=1 262 047 | Antipsychotics during pregnancy vs antipsychotics before/after pregnancy | Antipsychotics during pregnancy vs no antipsychotic use | Antipsychotics before/after pregnancy vs no antipsychotic use |
|---|---|---|---|---|---|---|
| | n (%) | n (%) | n (%) | P value | P value | P value |
| Perinatal death | 16 (0.6) | 162 (0.5) | 5057 (0.4) | 0.357 | 0.108 | 0.046 |
| Infant sex | | | | 0.777 | 0.610 | 0.454 |
| Male | 1365 (51.0) | 17 660 (51.2) | 644 906 (51.1) | | | |
| Female | 1312 (49.0) | 16 832 (48.8) | 617 141 (48.9) | | | |

*No use during the entire study period.
†Including promethazine (R06AD02, R06AD52), alimemazine (R06AD01), dixyrazine (N05AB01), prochlorperazine (N05AB04) and melperone (N05AD03).
BMI, body mass index.

The NNH for NICU admission compared with the unexposed infants adjusted for maternal factors was 18.

The neonatal disorders with increased risks associated with antipsychotics exposure were withdrawal symptoms from therapeutic drugs (aRR: 17.7, 95% CI: 9.6 to 32.6), neurological disorders (aRR: 3.4, 95% CI: 2.5 to 4.7) and PPHN (aRR: 2.1, 95% CI: 1.4 to 3.1) when compared with the unexposed infants. However, the absolute frequencies of these diagnoses were still less than 2% among the exposed infants (table 3). A sensitivity analysis on PPHN only including term infants born at gestational week 37 or later did not change the risk estimates. Adjusted NNH for

neurological disorders was 139 and for PPHN 227, when compared with the unexposed infants.

The most frequent outcomes affecting 5%–7% of the exposed infants were respiratory disorders, hyperbilirubinaemia and hypoglycaemia. The adjusted risks for these conditions were moderately increased for exposed infants, slightly lower when compared with the control group exposed before/after pregnancy than in comparison with the unexposed infants (table 3). Additional adjustment for fetal factors including prematurity did not markedly change these risks (online supplemental table 3). No statistically significant differences between the groups

**Table 2** Risk ratio (RR) for admission to neonatal intensive care unit (NICU) compared with non-exposed infants and with infants to mothers using antipsychotics before and/or after but not during the current pregnancy

| | NICU admissions | Exposed vs never exposed | | | | Exposed infants vs infants to mothers treated with antipsychotics before or after the pregnancy | | | |
|---|---|---|---|---|---|---|---|---|---|
| | n (%) | Crude RR | 95% CI | Adjusted RR | 95% CI | Crude RR | 95% CI | Adjusted RR | 95% CI |
| **Exposed early pregnancy only** | | | | | | | | | |
| Any antipsychotics, n=1454 | 251 (17.3) | 2.2 | 2.0 to 2.5 | 1.5 | 1.3 to 1.7 | 1.5 | 1.3 to 1.7 | 1.3 | 1.1 to 1.4 |
| First generation, n=431 | 76 (17.6) | 2.2 | 1.7 to 2.7 | 1.5 | 1.2 to 1.9 | 1.5 | 1.2 to 1.8 | 1.3 | 1.0 to 1.6 |
| Second generation, n=1166 | 212 (18.2) | 2.2 | 2.0 to 2.6 | 1.5 | 1.3 to 1.7 | 1.5 | 1.3 to 1.7 | 1.3 | 1.1 to 1.4 |
| **Exposed in late pregnancy** | | | | | | | | | |
| Any antipsychotics, n=1223 | 265 (21.7) | 2.8 | 2.5 to 3.1 | 1.8 | 1.6 to 2.0 | 1.9 | 1.7 to 2.1 | 1.5 | 1.4 to 1.7 |
| First generation, n=297 | 73 (24.6) | 3.1 | 2.6 to 3.8 | 2.1 | 1.7 to 2.5 | 2.1 | 1.7 to 2.6 | 1.7 | 1.4 to 2.1 |
| Second generation, n=972 | 208 (21.4) | 2.7 | 2.4 to 3.1 | 1.8 | 1.6 to 2.0 | 1.9 | 1.6 to 2.1 | 1.5 | 1.3 to 1.7 |
| Never exposed, n=1 262 047 | 98 976 (7.8) | Reference group 1 | | | | | | | |
| Exposed before or after but not during pregnancy, n=34 492 | 4010 (11.6) | Reference group 2 | | | | | | | |

Adjusted for: primipara, age, body mass index, smoking, caesarean section and concurrent neurotropic drugs.

**Table 3** Risk ratio (RR) for birth defects and neonatal morbidity after exposure to antipsychotics

| Infant outcomes | Exposed during pregnancy N=2677 n (%) | Exposed before/after but not during pregnancy N=34 492 n (%) | Never exposed N=1 262 047 n (%) | Exposed vs never exposed | | | | Exposed infants vs infants to mothers treated with antipsychotics before or after the pregnancy | | | |
| --- | --- | --- | --- | --- | --- | --- | --- | --- | --- | --- | --- |
| | | | | Crude RR | 95% CI | Adjusted RR | 95% CI | Crude RR | 95% CI | Adjusted RR | 95% CI |
| Any birth defect* | 64/2378 (2.7) | 27 122 (2.1) | 822 (2.4) | 1.3 | 1.0 to 1.6 | 1.1 | 0.8 to 1.4 | 1.1 | 0.9 to 1.5 | 1.1 | 0.8 to 1.4 |
| Any heart defect* | 23/2378 (1.0) | 9644 (0.8) | 304 (0.9) | 1.3 | 0.8 to 1.9 | 1.1 | 0.7 to 1.6 | 1.1 | 0.7 to 1.7 | 1.0 | 0.7 to 1.6 |
| Admission to NICU | 516 (19.3) | 4010 (11.6) | 98 976 (7.8) | 2.5 | 2.3 to 2.7 | 1.7 | 1.6 to 1.8 | 1.7 | 1.5 to 1.8 | 1.4 | 1.3 to 1.5 |
| Respiratory symptoms | | | | | | | | | | | |
| Transient tachypnoea of the newborn | 185 (6.9) | 1406 (4.1) | 34 204 (2.7) | 2.5 | 2.2 to 2.9 | 1.6 | 1.4 to 1.9 | 1.7 | 1.5 to 2.0 | 1.4 | 1.2 to 1.7 |
| Persistent pulmonary hypertension | 27 (1.0) | 209 (0.6) | 4581 (0.4) | 2.8 | 1.9 to 4.1 | 2.1 | 1.4 to 3.1 | 1.7 | 1.1 to 2.5 | 1.5 | 1.0 to 2.2 |
| Respiratory distress syndrome | 21 (0.8) | 296 (0.9) | 6671 (0.5) | 1.5 | 1.0 to 2.3 | 1.0 | 0.7 to 1.6 | 0.9 | 0.6 to 1.4 | 0.9 | 0.6 to 1.4 |
| Respiratory treatment | | | | | | | | | | | |
| CPAP | 168 (6.3) | 1276 (3.7) | 29 977 (2.4) | 2.7 | 2.3 to 3.1 | 1.7 | 1.4 to 2.0 | 1.7 | 1.5 to 2.0 | 1.5 | 1.2 to 1.7 |
| Ventilator treatment | 22 (0.8) | 299 (0.9) | 6811 (0.5) | 1.5 | 1.0 to 2.3 | 1.1 | 0.7 to 1.6 | 1.0 | 0.6 to 1.5 | 0.9 | 0.6 to 1.4 |
| Hyperbilirubinaemia | 169 (6.3) | 1819 (5.3) | 54 563 (4.3) | 1.5 | 1.3 to 1.7 | 1.3 | 1.1 to 1.5 | 1.2 | 1.0 to 1.4† | 1.1 | 1.0 to 1.3 |
| Hypoglycaemia | 133 (5.0) | 1090 (3.2) | 28 117 (2.2) | 2.2 | 1.9 to 2.6 | 1.4 | 1.2 to 1.7 | 1.6 | 1.3 to 1.9 | 1.3 | 1.1 to 1.5 |
| Feeding difficulties | 75 (2.8) | 558 (1.6) | 14 147 (1.1) | 2.5 | 2.0 to 3.1 | 1.8 | 1.4 to 2.3 | 1.7 | 1.4 to 2.2 | 1.6 | 1.2 to 2.0 |
| Neurological disorders | 47 (1.8) | 186 (0.5) | 3821 (0.3) | 5.8 | 4.4 to 7.7 | 3.4 | 2.5 to 4.7 | 3.3 | 2.4 to 4.5 | 2.3 | 1.6 to 3.3 |
| Withdrawal symptoms from therapeutic drugs | 34 (1.3) | 82 (0.2) | 153 (0.0) | 105 | 72.4 to 152 | 17.7 | 9.6 to 32.6 | 5.3 | 3.6 to 8.0 | 3.3 | 2.1 to 5.3 |

Adjusted for: primipara, age, body mass index, smoking, caesarean section and concurrent neurotropic drugs.
*Exposures are only counted if present in early pregnancy (n=2378).
†P<0.05.
CPAP, continuous positive airway pressure; NICU, neonatal intensive care unit.

**Table 4** Length of stay at neonatal intensive care unit (NICU) among term infants presented in median days and IQRs

| Length of stay among term infants admitted to NICU (days) | All term infants | | Infants with diagnosis of PPHN | | Infants with neurological disorders | |
|---|---|---|---|---|---|---|
| | n | Median (IQR) | n | Median (IQR) | n | Median (IQR) |
| Exposed | 336 | 5 (3–9) | 12 | 8 (3–9) | 34 | 9 (6–11) |
| Exposed before/after pregnancy | 2380 | 5 (2–8) | 80 | 9 (6–15) | 129 | 11 (7–18) |
| Not exposed | 59 439 | 5 (2–8) | 1894 | 9 (5–17) | 2698 | 11 (7–18) |

Presented for all term infants, infants with PPHN and infants with neurological disorder.
PPHN, persistent pulmonary hypertension.

were found for respiratory distress syndrome, ventilator treatment, any malformations or heart malformations.

To grade the severity of the neonatal morbidity after exposure to antipsychotics, length of stay was calculated for the full-term infants admitted to NICU (table 4). Median length of stay in infants exposed to antipsychotics during pregnancy was 5 days, similar to the infants not exposed. However, when corrected for maternal factors, the exposed infants had slightly but statistically significantly longer length of stay at NICU when compared with infants to mothers never exposed to antipsychotics. For full-term infants with PPHN and/or neurological disorders, the length of stay at NICU was slightly shorter than for the unexposed infants.

## DISCUSSION

Neonatal illness and the risk of admission to NICU were found to be increased among the exposed infants, at similar levels for both first-generation and second-generation antipsychotics. The risk of needing neonatal care was further increased after exposure in late pregnancy. The equal or slightly longer length of stay at NICU in exposed infants compared with never exposed infants indicates that these admissions reflect real neonatal illness and are not only a cautionary measure due to the exposure. The risk of PPHN was also clearly increased for the exposed infants, which has to our knowledge not been previously described related to antipsychotics.

Our results provide a detailed description of the neonatal morbidity that has not been published before in a study of this size. It is likely that the larger cohort size in this study enables studying the rarer neonatal outcomes including the two overlapping diagnoses of neurological outcomes and drug withdrawal symptoms. These outcomes have not previously been shown to be increased among exposed infants, but in this study, they are clearly increased even after adjustments for maternal factors.[22 23] An increase in neonatal morbidity after use of antipsychotics in late pregnancy that we found has also been observed for selective serotonin reuptake inhibitors (SSRIs).[36] Considering the mechanisms of these drugs in the serotonergic and dopaminergic pathways of the central nervous system, there is likely to be a biological,

drug-related explanation behind this.[37] However, the causality of this connection remains to be clarified.

The Swedish nationwide registers have some limitations regarding data on the drug exposure, which are based on filled prescriptions (PDR) and patient interviews (MBR) and not containing information on to what extent the women took the drugs. It is likely that some degree of exposure misclassification exists. Since information on drug intake was collected before the outcomes were known, it is unlikely the exposure misclassification has seriously biased the results. Even with these drawbacks, we consider the results from this study to be well generalisable to other cohorts, due to the nationwide heterogeneous cohort.

The risk estimates for NICU admission and the studied neonatal disorders decreased but remained significant when exposed infants were compared with infants to mothers treated with antipsychotics before or after pregnancy. This indicates that a part of, but not all, the risk increase for neonatal morbidity seen in the exposed infants is likely to be attributable to underlying factors rather than the drug treatment. However, as the results remained significant after adjustments, it is likely that there is also a true risk of neonatal morbidity connected to the antipsychotic drug treatment, especially after exposure in late pregnancy. Interestingly, the risk of being admitted to NICU was similar for infants to mothers with antipsychotic treatment in monotherapy and infants whose mothers had used other neurotropic drugs together with antipsychotics during pregnancy. This indicates that the connection between antipsychotic exposure during late pregnancy and admission to NICU cannot be explained by the high percentage of mothers with concomitant medications in the exposed group. For policymakers, it is important to consider the need for well-developed and structured healthcare for women with psychiatric disorders and their infants. For the clinicians, it is also important to acknowledge the crude risk estimates of this study, as they reflect the true risks for this group of patients, burdened with comorbidities. Whether there is a dose–response relationship to these effects remains to be studied.

The risk increase for PPHN that was found could be explained by poor adaptation in these infants due to the

neurological effects of the drugs, leading to both respiratory disorders and down the line even PPHN. Over half of the exposures in our data set consist of olanzapine and quetiapine, two substances affecting serotonergic as well as dopaminergic receptors.[38] A similar, but milder, correlation to PPHN has also been shown for prenatal exposure for SSRIs.[36 39] Therefore, we speculate that an effect on the serotonergic system might partly explain the increased risk of PPHN. The length of stay at NICU was shorter in exposed full-term infants with PPHN than in their unexposed peers, indicating that PPHN connected to fetal exposure to antipsychotics could be less severe than PPHN secondary to other causes. Some of the risk increase could also be due to overdiagnosis of PPHN among exposed infants with respiratory disorders. The risk of neurological outcomes seems increased on group level, but whether it is a causal effect of the antipsychotic drugs and what neurological outcomes these infants have increased risks for is yet to be determined. The length of stay at NICU was slightly shorter in infants exposed to antipsychotics with neurological disorders than unexposed infants, but still rather long, 9 days in median, indicating that even though some overdiagnosis of the exposed infants might be present, the exposed infants did require more than just mere observation. Further studies with larger cohorts are needed to separate the risks for the different antipsychotic drugs and the different neurological outcomes from each other.

## CONCLUSION

Fetal exposure to antipsychotics is associated with an increased risk of neonatal morbidity. The risk seems similar for infants exposed to first-generation and second-generation antipsychotics, and more pronounced after exposure in late pregnancy, and the increased risks of PPHN and neurological outcomes require further studies. In general, the neonatal disorders seen in exposed infants seem transient and predominantly mild, and these findings do not warrant discontinuation of a necessary treatment but rather increased monitoring of these infants post partum.

**Contributors** All authors planned and designed the study together and approved the final manuscript as submitted and agree to be accountable for all aspects of the work. KK was responsible for the data collection and acts as the guarantor of the work. EH drafted the initial manuscript. KK, LF, UN and KW reviewed and revised the manuscript.

**Funding** This work was supported by the Swedish Research Council (grant number 521-2012-3466; EH, KW, LF) and Stockholm County Council (ALF) (grant number 573301; EH, KW, LF).

**Disclaimer** The funders did not participate in the work.

**Competing interests** None declared.

**Patient and public involvement** Patients and/or the public were not involved in the design, or conduct, or reporting, or dissemination plans of this research.

**Patient consent for publication** Not required.

**Ethics approval** The study was approved by the regional ethical review board in Lund (dnr. 2019-02066).

**Provenance and peer review** Not commissioned; externally peer reviewed.

**Data availability statement** Data may be obtained from a third party and are not publicly available. Individual data are protected by the General Data Protection Regulation in EU (GDPR). All data are stored and accessible with permission only in a data repository at the Swedish National Board of Health and Welfare.

**ORCID iD**
Essi Heinonen http://orcid.org/0000-0003-2107-0561

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
