## [Reviewer comments · BMJ Open]

ARTICLE DETAILS

TITLE (PROVISIONAL)	Neonatal morbidity after fetal exposure to antipsychotics - a national register-based study
AUTHORS	Heinonen, Essi; Forsberg, Lisa; Nörby, Ulrika; Wide, Katarina; Källén, Karin

VERSION 1 – REVIEW

REVIEWER	Kaye, Adam University of the Pacific, Pharmacy Practice
REVIEW RETURNED	10-Feb-2022

GENERAL COMMENTS	Well done
-----------

REVIEWER	Payne, Jennifer Johns Hopkins Hospital and Health System
REVIEW RETURNED	20-Mar-2022

GENERAL COMMENTS	This is a well-written, thorough and well designed study examining the risk of in utero antipsychotic exposure. The study is well controlled and the discussion is thoughtful.
--

REVIEWER	Viguera, Adele Cleveland Clinic
REVIEW RETURNED	06-Apr-2022

GENERAL COMMENTS	The authors address an important clinical issue in their manuscript entitled, neonatal morbidity after fetal exposure to antipsychotics—a national register-based study. The aim of this study (based on the Swedish population register) is to clarify the pattern and frequency of the neonatal effects associated with in utero exposure to antipsychotics, primarily SGAs. A unique contribution of this analysis is also the attempt to separate these effects from the impact of the underlying disease. They achieve this by including a reference group with women exposed to antipsychotic any time during the study period, before or after pregnancy but not during pregnant or 1 month before it. Overall, this is a very well written manuscript and clinically significant. Only minor revisions are required. General comments: 1. The authors should address in their discussion their observation that for full term infants with PPHN and/or neurological disorders,
---

	their length of stay in the NICU was slightly shorter than for the unexposed infants (page 9 lines 49-52). This finding seems curious and counterintuitive. Can the authors hypothesize on why this may be true? 2. The authors comment “ The risk for PPHN was also clearly increased for the exposed infants, which was to our knowledge not been previously described related to antipsychotics”. The authors should elaborate further in the discussion – why do they think they are observing this association when it has not been observed in other studies? How is PPHN being defined in the Swedish records? Is there a severity scale? One would think that infants with PPHN would have prolonged hospital stays but that was not observed. Then are these just mild cases with transient symptoms that resolve spontaneously or do they require serious intervention? In clinical practice in perinatal psychiatry, one very rarely sees PPHN (if at all) associated with SSRIs or for that matter with second generation antipsychotics. However, in the literature such an association between SSRIs and PPHN has been observed and unfortunately grabbed headlines, probably causing more harm than good by scaring patients and clinicians into discontinuing medication before or at delivery to reduce this risk. Therefore, It would be important to emphasize in the discussion (as the authors stated in the abstract) the absolute risks for such an outcome “were however very low” and elaborate further on the clinical context and implications. 3. The authors in the discussion state “The neonatal disorders seem transient and predominately mild, and these findings do not warrant discontinuation of a necessary treatment but rather an increased monitoring of infants postpartum”. As discussed above, perhaps the authors should consider applying this qualifier to the association of antipsychotic exposure and PPHN to provide necessary clinical context. 4. It would seem the PPHN would be associated with a greater length of stay in the NICU but the authors observed the contrary (see comment #1). Is it possible that somehow the diagnosis of PPHN is capturing very mild form of the disease? 5. Include in the methods with greater clarity the use of concomitant psychotropic. What was the frequency of polytherapy vs monotherapy (i.e. psychotropics) overall in the sample (not broken down by class of medication) ? Did there appear to be a dose response relationship with the greater the number of concomitant medications , the greater the risk for infant morbidity ? for PPHN? etc
--	--

VERSION 1 – AUTHOR RESPONSE

Reviewer Reports:

Reviewer 1: Prof. Adam Kaye, University of the Pacific

Comments to the Author:

Well done

Thank you so much!

Reviewer 2: Prof. Jennifer Payne, Johns Hopkins Hospital and Health System

Comments to the Author:

This is a well-written, thorough and well-designed study examining the risk of in utero antipsychotic exposure. The study is well controlled, and the discussion is thoughtful.

Thank you! We are very honored by this positive feed-back!

Reviewer 3: Dr. Adele Viguera, Cleveland Clinic

Comments to the Author:

The authors address an important clinical issue in their manuscript entitled, neonatal morbidity after fetal exposure to antipsychotics—a national register-based study. The aim of this study (based on the Swedish population register) is to clarify the pattern and frequency of the neonatal effects associated with in utero exposure to antipsychotics, primarily SGAs. A unique contribution of this analysis is also the attempt to separate these effects from the impact of the underlying disease. They achieve this by including a reference group with women exposed to antipsychotic any time during the study period, before or after pregnancy but not during pregnant or 1 month before it. Overall, this is a very well written manuscript and clinically significant. Only minor revisions are required.

General comments:

1. *The authors should address in their discussion their observation that for full term infants with PPHN and/or neurological disorders, their length of stay in the NICU was slightly shorter than for the unexposed infants (page 9 lines 49-52). This finding seems curious and counterintuitive. Can the authors hypothesize on why this may be true?*

Thank you for this observant comment! We have some theories for these differences, yes. Regarding PPHN we hypothesize, that full term infants acquiring PPHN of other reasons might have other underlying factors, such as structural lung disorders (e.g. cystic lung malformations, diaphragmatic hernia, lung dysplasia) or asphyxia predisposing to PPHN. These underlying disorders themselves often require long stays at NICU, so the PPHN caused by “only” what we hypothesize is an adaptational disorder connected to the drug exposure is less severe in comparison. Also, there might be a detection bias hidden in the results, as the first quantile of length of stay for exposed infants was only 3 days (compared to 5-6 days in the other groups), which is a very short time to be admitted to NICU if you have a fulminant PPHN. Therefore, we suspect that the threshold for performing a cardiac ultrasound might be lower for clinicians when an infant exposed to antipsychotics presents with respiratory disorder than for other infants with milder respiratory disorders.

Regarding the length of stay at NICU for infants with neurological outcomes, we also believe a detection bias might be a part of the explanation. We believe that the threshold for admitting an infant presenting with neurological symptoms to NICU is lower if the infant has been exposed to antipsychotics. However, the median length of stay in exposed infants is still 9 days (interquartile range 6-11 days), indicating that the neurological disorders presenting in exposed infants were still fairly severe.

We have now added a discussion explaining these hypotheses in short, section 4 of the discussion, pages 12-13.

2. *The authors comment “ The risk for PPHN was also clearly increased for the exposed infants, which was to our knowledge not been previously described related to antipsychotics”. The authors should elaborate further in the discussion – why do they think they are observing this association when it has not been observed in other studies? How is PPHN being defined in the Swedish records? Is there a severity scale? One would think that infants with PPHN would have prolonged hospital stays but that was not observed. Then are these just mild cases with transient symptoms that resolve spontaneously, or do they require serious intervention? In clinical practice in perinatal psychiatry, one very rarely sees PPHN (if at all) associated with SSRIs or for that matter with second generation antipsychotics. However, in the literature such an association between SSRIs and PPHN has been observed and unfortunately grabbed headlines, probably causing more harm than good by scaring patients and clinicians into discontinuing medication before or at delivery to reduce this risk. Therefore, It would be important to emphasize in the discussion (as the authors stated in the abstract) the absolute risks for such an outcome “were however very low” and elaborate further on the clinical context and implications.*

Thank you for bringing the perinatal psychiatric point of view to the discussion!

As the answer to the first part of the question why others have not found this connection before: Firstly, not all registers include the individual neonatal diagnoses and most register-based studies have therefore not had the option to study PPHN before. Prospective studies and smaller cohorts again have most likely not found the connection simply because the absolute risk for PPHN is still as low as 1% amongst exposed and 0.4% in unexposed infants, requiring huge cohort sizes to show a statistically significant difference between the groups.

And as we discussed in comment #1, yes, some over-diagnostics of PPHN is probably likely amongst the exposed infants. PPHN is only diagnosed with cardiac ultrasound, and its symptoms of requiring CPAP and oxygen are fairly similar to the ones of transient tachypnea of newborn. As cardiac ultrasound is not performed in all infants that require treatment with CPAP for a few days, it is likely that the knowledge of the exposure has lowered the threshold for the clinician to perform a cardiac ultrasound and therefore increasing the chance of the infant acquiring the diagnosis of PPHN, compared to if the ultrasound was not performed. The severity of the PPHN is not possible to study further than comparing the length of stay at NICU due to the rare exposure and small sample of exposed infants with PPHN (27 infants).

We now clarified in section 4 of the discussion, page 12, to increase the transparency further: *“The length of stay at NICU was shorter in exposed full-term infants with PPHN than in their unexposed peers, indicating that PPHN connected to fetal exposure to antipsychotics could be less severe than PPHN secondary to other causes. Some of the risk-increase could be due to over-diagnostics of PPHN amongst exposed infants with respiratory disorders.”*

3. *The authors in the discussion state “The neonatal disorders seem transient and predominately mild, and these findings do not warrant discontinuation of a necessary treatment but rather an increased monitoring of infants postpartum”. As discussed above, perhaps the authors should consider applying this qualifier to the association of antipsychotic exposure and PPHN to provide necessary clinical context.*

Thank you for this comment! We agree that the conclusion as it was stated previously could leave some questions. We have re-phrased the conclusion now to hopefully tone it down a notch and moved the qualifier to the end of the conclusion for more attention. The conclusion

now states: *“Fetal exposure to antipsychotics is associated with an increased risk for neonatal morbidity. The risk seems similar for infants exposed to first- and second- generation antipsychotics, and more pronounced after exposure in late pregnancy, and the increased risks for PPHN and neurological outcomes require further studies. In general, the neonatal disorders in the exposed infants seem transient and predominantly mild, and these findings do not warrant discontinuation of a necessary treatment but rather increased monitoring of these infants postpartum.”*

4. *It would seem the PPHN would be associated with a greater length of stay in the NICU but the authors observed the contrary (see comment #1). Is it possible that somehow the diagnosis of PPHN is capturing very mild form of the disease?*

Very true, the length of stay at neonatology is rather long for all infants regardless of exposure status (8 vs 9 days in median). However, as discussed in comment #1, PPHN can also be caused by several other more serious causes like asphyxia and lung malformations, causing the length of stay for exposed infants with PPHN connected to the drug exposure being relatively shorter. As we discussed in comment #1 and added to section 4 of the discussion on page 12, it is also likely as you state, that some infants with a milder form of PPHN are caught amongst the exposed infants due to relative over-diagnostics of exposed infants with respiratory disorders with cardiac ultrasound compared to unexposed infants, with length of stays as short as 3 days (the first quartile of the length of stay of exposed infants with PPHN) seen in exposed infants, which is rarely the case for fulminant PPHN.

Thank you for pointing this out and we hope that we have now clarified the discussion enough! However, the question of PPHN is rather complex, and as we state, more studies are needed to confirm the causality of PPHN and to understand the pathomechanism and the potential long-term effects caused by the PPHN for exposed the infants.

5. *Include in the methods with greater clarity the use of concomitant psychotropic. What was the frequency of polytherapy vs monotherapy (i.e. psychotropics) overall in the sample (not broken down by class of medication)? Did there appear to be a dose response relationship with the greater the number of concomitant medications , the greater the risk for infant morbidity ? for PPHN? Etc.*

Thank you for this comment!

We agree that as complicated studying the effects of polypharmacy is, it is also utterly important. With that said, it is a rather complicated question to answer in a large register-based setting like ours.

Regarding the methods section, we have now clarified the exposure for concomitant drugs in the methods-section, thank you for noticing the lack of definition of concomitant drug use! We have now added the following line to the end of section 4 in the methods-section, page 6:

“Exposure data was also collected on the following neurotropic drugs known to or suspected to cause similar neonatal morbidity as antipsychotics: antidepressants (ATC-code N06A), antiepileptics (N03A), opioids (N02A), centrally acting sympathomimetics (N06BA), sedatives (N05B, N05C) and milder sedatives (alimemazine, promethazine and the excluded antiemetic antipsychotics from N05A). Concomitant neurotropic drug exposure was defined as a prescription of these drugs dispensed at any time during or 1 month before the pregnancy and was adjusted for in the multivariable regression model.”

In the exposed group, 40.7% of the mothers used any other neurotropic drug during pregnancy, compared to 2.7% in the unexposed group. We have clarified this in section 2 of the results section, page 9. Originally, we addressed the question of polypharmacy with adjusting for the effects of these drugs in the multivariable regression analysis. For the revised version, we also made sensitivity analyses, evaluating any use of antipsychotics in late pregnancy in relation to admission to NICU and the occurrence of PPHN, respectively, in which we excluded women who had used any concomitant psychotropic drugs. The point estimates were quite similar to those obtained after adjustment for concomitant psychotropics. Furthermore, interestingly enough, the point estimates for the mentioned outcomes after antipsychotic exposure in late pregnancy among those with concomitant use of other psychotropics were similar to those without such concomitant treatment, as we have now added to the results-section, section 3, page 9, and discussed in section 4 of the discussion, page 12.

Thank you again for the positive and constructive comments, from both the editor and the reviewers, especially dr. Viguera! We are grateful and believe that the quality of the manuscript has increased thanks to these observant comments!

With best regards, Essi Heinonen and the rest of the study group.